# HF-SCA: Hands-Free Strong Customer Authentication Based on a Memory-Guided Attention Mechanisms

**Cosimo Distante** [1,†] **, Laura Fineo** [2,†] **, Luca Mainetti** [3,†] **, Luigi Manco** [4,†] **, Benito Taccardi** [5,†]
**and Roberto Vergallo** [3,*,†]

1   Institute of Applied Sciences and Intelligent Systems (ISASI), National Research Council of Italy,
    73100 Lecce, Italy; cosimo.distante@cnr.it
2   Department of Marketing, Banca Sella S.p.A., 13900 Biella, Italy; laura.fineo@sella.it
3   Department of Innovation Engineering, University of Salento, 73100 Lecce, Italy; luca.mainetti@unisalento.it
4   Vidyasoft s.r.l., 73047 Monteroni di Lecce, Italy; luigi.manco@vidyasoft.it
5   Faculty of Engineering, University of Salento, 73100 Lecce, Italy; benito.taccardi@studenti.unisalento.it
*   Correspondence: roberto.vergallo@unisalento.it; Tel.: +39-0832-297376
†   These authors contributed equally to this work.

**Abstract:** Strong customer authentication (SCA) is a requirement of the European Union Revised Directive on Payment Services (PSD2) which ensures that electronic payments are performed with multifactor authentication. While increasing the security of electronic payments, the SCA impacted seriously on the shopping carts abandonment: an Italian bank computed that 22% of online purchases in the first semester of 2021 did not complete because of problems with the SCA. Luckily, the PSD2 allows the use of transaction risk analysis tool to exempt the SCA process. In this paper, we propose an unsupervised novel combination of existing machine learning techniques able to determine if a purchase is typical or not for a specific customer, so that in the case of a typical purchase the SCA could be exempted. We modified a well-known architecture (U-net) by replacing convolutional blocks with squeeze-and-excitation blocks. After that, a memory network was added in a latent space and an attention mechanism was introduced in the decoding side of the network. The proposed solution was able to detect nontypical purchases by creating temporal correlations between transactions. The network achieved 97.7% of AUC score over a well-known dataset retrieved online. By using this approach, we found that 98% of purchases could be executed by securely exempting the SCA, while shortening the customer's journey and providing an elevated user experience. As an additional validation, we developed an Alexa skill for Amazon smart glasses which allows a user to shop and pay online by merely using vocal interaction, leaving the hands free to perform other activities, for example driving a car.

**Keywords:** strong customer authentication; transaction risk analysis; risk-based assessment; PSD2; machine learning; user experience; vocal interaction

## 1. Introduction

Most applications have client-side requirements that fall both in the security and user experience (UX) areas. However, while for non-mission-critical applications, authentication can be performed once, in digital payments, security and UX are at the extremities of a too-short blanket: you cannot settle payments without disturbing the user, and you cannot deliver a smooth UX without posing a risk to the customer. Beginning from 1 January 2021, strong customer authentication (SCA) has become mandatory for every digital purchase made within the European Economic Area (EEA) as per the mandate of the European Revised Directive on Payment Services (PSD2). The SCA process forces the customer to use a second factor in order to complete the payment; for example, the customer must open a push notification on their mobile phone, or copy in a web page a long code received via SMS, or identify themselves with a biometric technology, after the insertion of card data.

The consequences have been relevant for the banking industry: an Italian bank computed from its databases that a huge 22% of online purchases in the first semester of 2021 did not complete because of problems with the SCA. Reasonably, the SCA impacted mostly on the purchase made by the less accustomed to technology: for such people (e.g., older generations Reshetnikova et al. (2021); Imam et al. (2022), but also people who simply cannot afford a second device) paying online has become suddenly more difficult or even impossible. Excluding such a big part of citizens from the technological progress of the smart city goes exactly in the opposite direction of the aims of the smart city itself: citizen engagement, here including also the financial inclusiveness Ebong and George (2021), is a strong pillar of the smart city manifesto, which calls for creating and fostering accessible urban services for everyone. An urgent call to action in this topic is demanded also by the UN, which in the Sustainable Development Goals (SDG) list (target 1.4) requires that by 2030 all the countries shall "ensure that all men and women, in particular the poor and the vulnerable, have equal rights to economic resources, as well as access to [...] appropriate new technology and financial services".

Luckily, the SCA mandate is complemented by some limited exemptions that aim to support a frictionless UX when transaction risk is low. SCA exemptions are defined based on the level of risk, amount, recurrence and the payment channel used for the execution of the payment. One of these exemptions falls under the transaction risk analysis (TRA) umbrella and says that SCA is not mandated where a payment service provider (PSP), having in place effective risk analysis tools, assesses that the fraud risk associated with a remote payment transaction is low. Leveraging SCA exemption is the key to address the financial inclusiveness problem in digital payments, and is considered the "perfect storm" for Machine learning (ML). Particularly, the application of anomaly detection (AD) techniques in TRA is supposed to bring significant advantages in balancing the security–UX trade-off, as it allows one to detect transactions in which the SCA process could be securely exempted. AD techniques allow one to detect data which significantly differ from the majority of instances. AD in the financial context is aimed at detecting frauds instead of anomalies. An anomaly is different from a fraud in that the former is an unusual and possibly erroneous observation that does not follow the general pattern of a drawn population, while the latter is an intentional deceptive action perpetuated against a firm for financial gains. The main idea in this paper is to mandate the use of SCA only when strictly needed, i.e., only when an atypical purchase behaviour for the customer (i.e., a probable fraud) is detected.

The AD topic is very challenging from different point of views. If our AD task aims at detecting frauds or intrusions, enemies could shape their behaviour in order to overcome the AD logic. Into an evolutionary domain in which normal actions change over time, normal data could be difficult to be recognised and also there are few datasets which present labelled data.

Due to the previous reasons, it is not so easy to solve AD problems, but in the wide field of ML, recently deep learning has been growing sharply and it has been achieving unprecedented results Chalapathy and Chawla (2019).

In this paper, a novel approach to AD in a financial context is presented. It consists of an original combination of existing ML techniques, having as a backbone a neural network based on a U-net. Assuming that the information gathered by a single transaction could not be sufficient to state if it is fraud or not, the backbone architecture has been augmented by a memory network. In addition to this, an attention mechanism has been introduced in order to improve the network's separation ability.

Our AD algorithm is an enabling technology for the smart city. Particularly, one of the priorities in smart cities is to increase the efficiency of smart mobility systems through the optimal use of digital technologies. In terms of modern technologies, a better use of networks and their safer and more reliable operation is a key factor to serve such main goal Kadłubek et al. (2022). As a proof for this, we implement a real-world usage scenario in which a customer performs online purchases by using an Alexa voice skill while driving.

Such scenario has been selected since it perfectly depicts the context for the final customer of the system, that is the user who makes online card-not-present transactions. We show that, for purchases that are considered "typical" for the specific customer, the SCA exemption is successfully triggered, and the customer is not disturbed during their journey in the car. Particularly, the vocal interface allows the user to complete payments completely hands-free, hence the name of our work "Hands-Free SCA".

The novelty of this paper is twofold: (1) to the best of our knowledge, the combination of ML techniques that we present provides outstanding performance not present in any other research work in the AD field using the same public dataset; (2) the case study proves that our algorithm enables the creation of novel payment use cases for the smart city.

The paper is structured as follows. Section 2 provides an in-depth overview of literature projects and their state of the art on fraud detection and more generally AD. Section 3 describes the materials and methods we used in our work, including the proposed architecture, and explains each constituting block. Section 4 discusses the algorithm's experimental results, as well as describing the case study implemented to validate the use of the algorithm for enabling novel payment experiences. Section 5 presents a discussion of the main findings. In conclusion, Section 6 sums up the main findings and drafts the conclusions.

This work has been developed with the support of Banca Sella S.p.A.—an Italian private bank which performed a technical and business validation for this algorithm— and Vidyasoft s.r.l.—a spin-off of Salento University which proposed and prototyped the algorithm during a contest called Sella Data Challenge Sella (n.d.).

## 2. Literature Review

The PSD2 directive provides the current legal foundation for electronic payments within the European Union (EU) EU (2016); Tapia Hermida (2018). The directive sets out rules concerning:

- A licensing regime for payment institutions;
- The transparency of conditions and information requirements for payment services, including charges;
- The rights and obligations of users and providers of payment services;
- Strict security requirements for electronic payments and the protection of consumers' financial data, in order to guarantee safe authentication and reduce the risk of fraud.

The last point is crucial with regards to the presented study, since it lays out the legislative framework that electronic transaction systems must implement and the liability in security for stakeholders involved in payment transactions.

In such context, PSD2 forces payment service providers to apply a strong customer authentication (SCA) when a payer initiates an electronic payment transaction, leading to a redesign of the customer's journey alongside an enhanced security in digital transactions, as deeply analysed in Paul et al. (2020); Saarnilehto (2018).

As introduced in the previous section, the issues related to a mandatory SCA application could be addressed by means of anomaly detection (AD) techniques. AD is faced up using different methods, from pure statistical approaches to novel deep learning techniques. In Table 1, some DAD (deep anomaly detection) solutions are reported.

As González et al. (2020) states about generative models for anomaly detection: "Generative models represent a promising approach for network anomaly detection, especially when considering the complexity and ever-growing number of time-series to monitor in operational networks". On the other hand, Higa et al. (2019) states that discriminative models use convolutional neural networks for extracting features in order to distinguish between normal and anomaly transactions. Hybrid methods combine the previous model architectures to exploit the positive sides of generative and discriminative approaches.

**Table 1.** Some common AD approaches.

| Techniques | Model Architecture | References |
|---|---|---|
| Generative | DCA [1], SAE [2], RBM [3], DBN [4], CVAE [5] | Yu et al. (2017); Zolotukhin et al. (2016); Garcia Cordero et al. (2016); Alrawashdeh and Purdy (2016); Tang et al. (2016); Lopez-Martin et al. (2017); Al-Qatf et al. (2018); Mirsky et al. (2018); Aygun and Yavuz (2017) |
| Hybrid | GAN [6] | Lin et al. (2021); Yin et al. (2018); Ring et al. (2019); Latah (2018); Intrator et al. (2018); Matsubara et al. (2018); Cao et al. (2016); Rigaki and Elragal (2017) |
| Discriminative | RNN [7], LSTM [8], CNN [9] | Malaiya et al. (2018); Althubiti et al. (2018); Naseer et al. (2018) |

[1] DCA is the acronym of Dominant Component Analysis. [2] SAE is the acronym for stacked autoencoder [3] RBM is the acronym for restricted Boltzmann machine [4] DBN is the acronym for deep belief network [5] CVAE is the acronym for conditional variational autoencoder [6] GAN is the acronym for generative adversarial network [7] RNN is the acronym for recurrent neural network [8] LSTM is the acronym for long short-term memory [9] CNN is the acronym for convolutional neural network.

AD is used in different applications from intrusion detection systems in a network as Zoppi et al. (2020) reports, to denial-of-service attack as explained by Ahmed (2017) and also in financial frauds as Cheng et al. (2020) states.

Asha and Kumar (2021) tries two different approaches to solve the problem of AD in credit card spending: Bayesian network and artificial neural network (ANN).

Starting from the first method, a Bayesian network is called in the literature a DAG (direct acyclic graph), that is, a data structure, more specifically, an oriented graph which contains no closed loops. Each edge is a conditional probability, and each node is a random variable. In addition to this approach, Asha and Kumar (2021) also tried another approach: feedforward multilayer perceptron. At the time when the cited paper was published it represented a novel solution to credit card fraud detection problem; nowadays, this kind of ANN is an entry-level neural network architecture.

During the last years, ANNs has played a main role in AI growth; for this reason, ANN architectures have become more and more complex and fancy. Therefore, convolutional neural networks (CNN) were created. These networks are based on series of convolutional operations. As Fu et al. (2016) stated, CNNs are suitable for several applications, especially for tasks involving images with some mechanisms (such as dropout, dataset augmentation, drop connections, etc.). In Fu et al. (2016), the authors showed a CNN designed to support fraud detection. As the paper asserted, the proposed architecture was very similar to the LeNet architecture, which was the first CNN, proposed by Rawat and Wang (2017).

Another approach to financial fraud AD was proposed by Randhawa et al. (2018). In that paper, the authors tried to solve the AD problem using AdaBoost and majority voting approaches.

AdaBoost (which is the contraction of adaptive boost) is not properly a novel algorithm but it is a way of evaluating a set of results and improving performance by exploiting the strength of each result. As the authors of the previous cited paper claimed, AdaBoost was used in conjunction with different types of algorithms to improve their performance. AdaBoost was able to adjust the classifier in order to better predict misclassified items. Nevertheless, adaptive boosting is quite sensitive to noise and outliers.

On the other hand, majority voting, like AdaBoost, is an algorithm which combines the results of two or more classifiers. Namely, majority voting outputs the class which received the majority of votes.

Moving towards more recent approaches, Jurgovsky et al. (2018) used recurrent neural networks (RNNs), specifically a long short-term memory (LSTM) network, which is a special case of an RNN with memory cells. As the authors claimed, by using connections

across time steps, the model can retain information about the past inputs, allowing it to discover temporal correlations between events that are possibly far away from each other in the input sequence. This is a crucial property for the proper learning of time series where the occurrence of one event might depend on the presence of several other events back in time. In fact, Jurgovsky et al. (2018) exploited the key feature of LSTM: holding information and seeking patterns to enable time correlations.

An RNN is able to remember information. For this reason, decisions taken during the evaluation process are heavily influenced by what the NN has learnt in the past, then using a temporal reference. To be more precise, feed forward neural networks also have a kind of memory, but their memory is constrained to what the training phase shows to the network. An RNN benefits from both references: training information and temporal information. However, the paper used a special case of an RNN, the LSTM network. This type of network solves a problem introduced by recurrent networks: short-term memory. In fact, some information could be left out from the beginning using an RNN. An LSTM network implements an internal mechanism in order to improve the memory skill of the network by unrolling the time. In Cheng et al. (2020), the authors proposed a novel approach based on attention networks. Attention mechanisms were introduced to improve memory networks (such as RNNs presented before). This solution was inspired by human behaviours for some actions. For example, while we drive our car, our focus is mainly on cars which surround the driver and not on the background of the scene. Our visual system and our brain are able to focus most of our attention to specific subjects of the whole scenario.

Several attention networks have been applied to sequence-to-sequence (Seq2Seq) models (examples of Seq2Seq models are autoencoders).

Attention mechanisms belong to two main categories: (1) Bahdanau attention, called also "additive attention", which is based on Bahdanau et al. (2016); (2) Luong attention, also called "multiplicative attention", which is based mainly on Luong et al. (2015).

The authors of Cheng et al. (2020) based their solution on this approach.

Due to its working process, autoencoders (AE) work quite well for AD tasks. For this reason, AE architectures are widely applied in anomaly detection tasks, but also for outlier recognition using images, sounds, signals and so on.

Chen et al. (2021) proposed several solutions to antilaundering or fraud detection based on deep-learning approach. In particular, the authors proposed two solutions based on a standard autoencoder architecture but also a solution based on a variational autoencoder (VAE) architecture. This novel approach is a stochastic version of a standard AE. In fact, a VAE tends to describe the latent space in a probabilistic way. For these reasons, the latent space being a continuous space, operations such as random sampling or interpolation are easier to perform.

The last paper reported in this review is one of the most efficient solutions for AD in video applications. In Park et al. (2020), a video AD was performed using architectures and mechanisms which were quite similar to those presented before. In particular, the authors of this solution used a neural network based on U-net. Ronneberger et al. (2015) presented this architecture for the first time and it was used mainly for image segmentation. In Figure 1, the U-net architecture is depicted. U-net got its name due to its shape, which resembles the letter "U".

In this image, different colours can be identified. Blue-coloured blocks represent regular convolutional layers. Purple blocks, on the other hand, represent dense blocks. These blocks instantiate all possible connections between neurons of two layers (with matching feature map) as Huang et al. (2016) explains. Moreover, arrows have different colours. Black arrows represent regular convolutional operations. Red arrows represent max pooling operations. Yellow arrows represent the connections of a dense layer; this means that there is one connection per couple of neurons between two connected layers. Finally, grey connections represent skip connections. The pictures report also some numbers. Those that are placed beside a block represent the dimension of the block itself in terms of

width and height of the feature map. On the other hand, those that are place above a block represents the depth of the feature map.

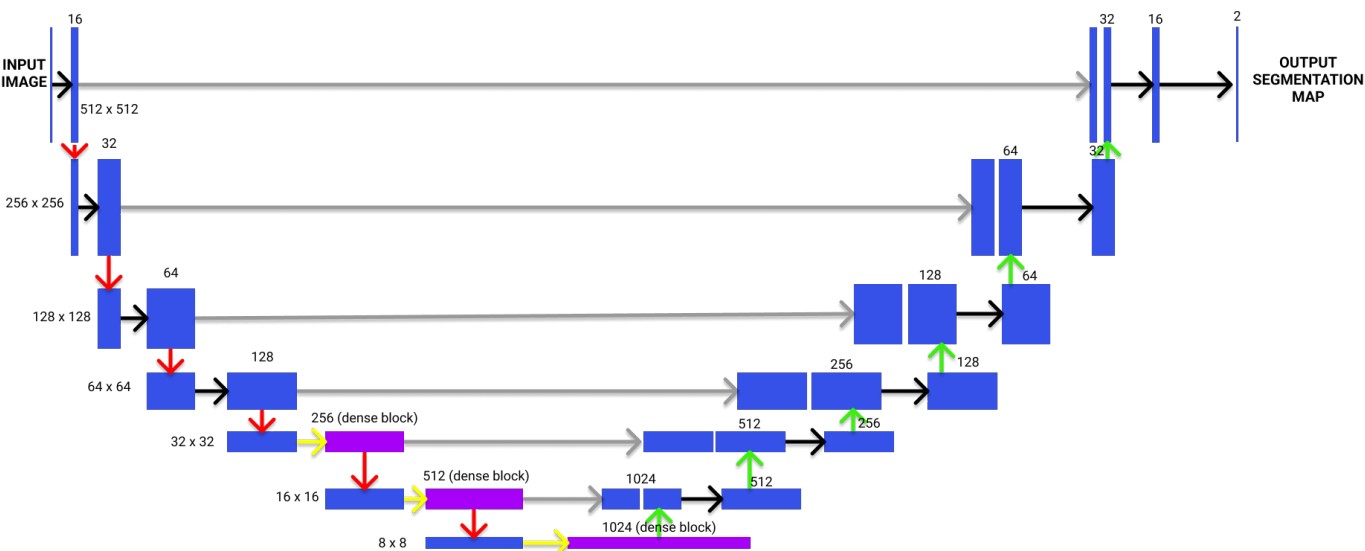

**Figure 1.** U-net architecture.

While fairly similar to a standard AE, U-net implements special connections (called "skip connections") allowing this architecture to avoid the bottleneck problem, so a high number of features is allowed to pass from the encoding to the decoding part. Skip connections link layers of the encoding side to equal levels of the decoding counterpart.

In addition to this, the authors of Park et al. (2020) placed in the latent space of the U-net architecture a memory network containing 10 memories. In this way, the whole architecture was able to reconstruct the signal not only using the latent-space information but also using memory-network information. MemNet proposed by Park et al. (2020) is also very important because it allows the system to reconstruct patterns which repeat themselves in a quite long period of time.

Fraud detection, as other AD problems, faces two main challenges: data imbalance and concept drift. The first means that frauds are fewer than normal transactions; the second problem means that customers' behaviours evolve. For these reasons, classical approaches which use machine learning algorithms could affect the accuracy of the results. As Gao et al. (2019) stated in their conclusions, random forests and isolation forests algorithms perform well in quantitative description of anomalies and from the computational point of view; on the other hand, they struggle in local anomaly points detection, which is a drawback that should be overtaken. CNNs represent a breakthrough with respect to classical machine learning approaches. Despite their improvements, CNNs are mainly applied in tasks which involve images, since they are able to reconstruct pixels' neighbourhood. Applying CNNs to fraud detection problems requires a new definition of "convolution" which could be hard to find. The state of the art has proposed AEs as a method to address AD problems. Reconstructing and comparing approaches succeed in anomaly identification, nevertheless, basic AEs cannot perform well in complex anomalous situations. In order to deal with problem complexity, memory-based approaches have demonstrated (not only in fraud detection tasks, but also in image AD) that they are a robust solution for tackling the problem from a multimodal point of view. The solution we propose in this paper leverages the memory-based benefits, and also strengthen the whole system using some neural network tools which improve the fraud detection ability of the network.

## 3. Materials and Methods

The architecture developed in this work is made up of different parts and was designed by taking inspiration from Park et al. (2020). Nevertheless, the model and components are clearly different. What is similar is the use of a memory network in a latent space.

Figure 2 shows an overall view of the architecture.

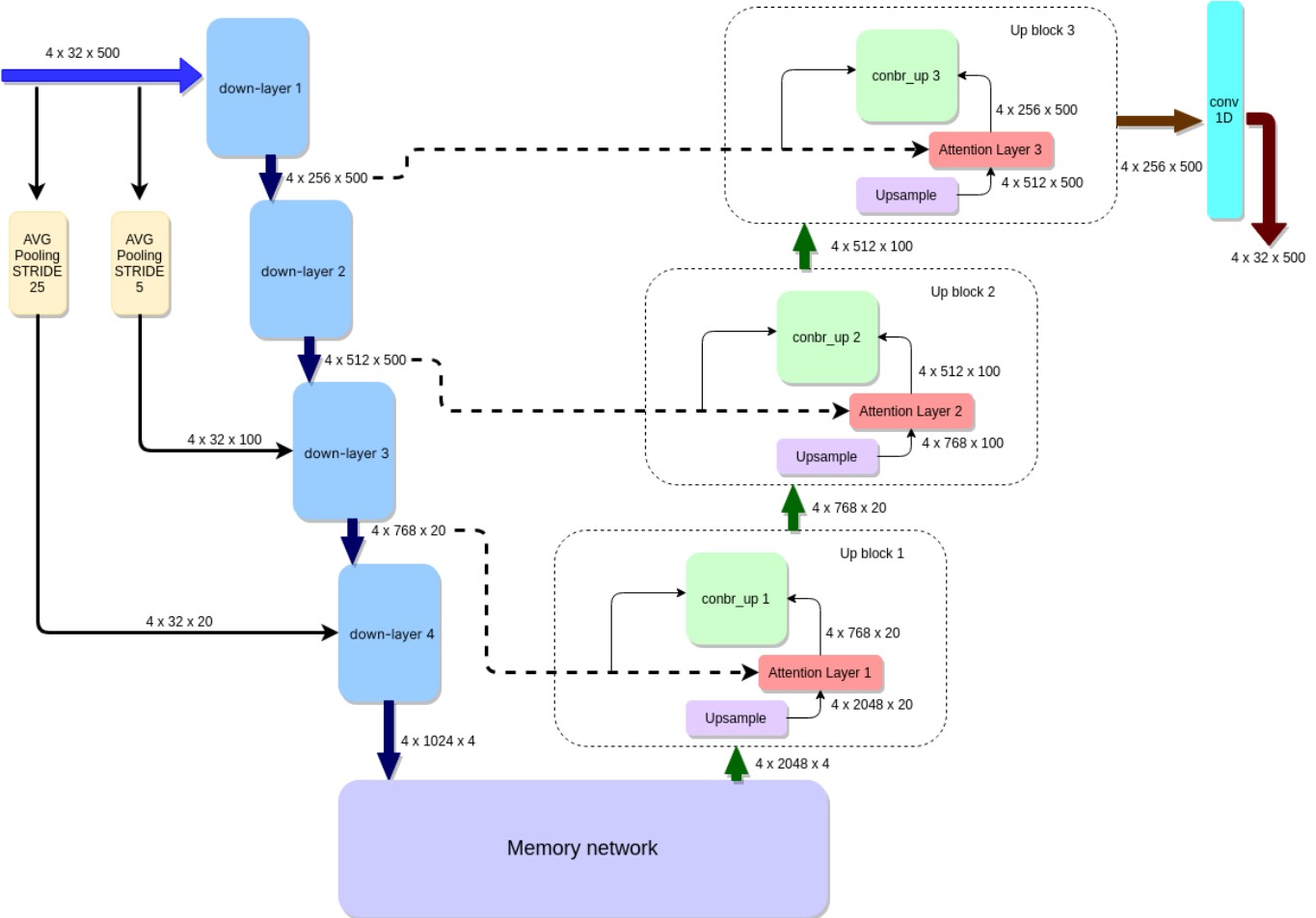

**Figure 2.** Overview of the proposed AD architecture proposed in this work.

Referring to the previous picture, each blue blocks represents a down-layer. It is composed of several atomic and nonatomic sub-blocks which perform information compression. Particularly, a down-layer is a sequence of a conbr-block and a re-block.

In addition to the macroblocks, Figure 2 also shows the input and output sizes of each layer. Dimensionalities are reported as batch size × number_features × sliding_window. Down-layers are composed by elementary blocks: convolutional, batch-normalization, ReLu, squeeze-and-excitation and sigmoid blocks. On the decoding side, the macro-block is depicted as a group of three main blocks: up-sample, contraction and attention blocks. The bottleneck layer is substituted by a memory network which enables a multimodal analysis of anomalies. The memory network is made up of an update part, which refreshes keys value in memory, and a reading part, which extracts the closest (or the k-closest) keys in the network.

Figure 2 represents just the global idea of the whole concept: each block in the figure is not atomic but rather contains several elementary blocks. For this reason, each block is explained in detail in the following sections.

The backbone of the architecture is a U-net architecture Ronneberger et al. (2015). We chose U-net because it enabled us to perform anomaly detection using a reconstruction-error-based method in a similar way to autoencoders. However, due to the deep levels of

the network itself, skip connections allowed us to avoid gradient-vanishing effects, thus having better performance during training and evaluation phases. Additionally, we clearly modified the U-net architecture using squeeze-and-excitation blocks Hu et al. (2019) instead of nonconvolutional blocks. In the latent space of the network, after the first encoding side, the latent features become input of a memory network, which is quite similar to the one proposed in Park et al. (2020). In the encoding phase, the "up block" is made up of several smaller blocks. An upsample block has the aim of augmenting the number of descriptors. Particularly, the main goal of the upsampling level is to bring back the resolution of the input to the resolution of the downsampling block of the encoding counterpart. The upsampling block is not the only one constituting the decoding block: it also contains an attention layer and a "conbr-up" block which is discussed in detail later.

Furthermore, when the original data input comes, two average pooling blocks with different strides are processed. Their output is used during the downsampling phase in order to restore some key features surviving the average pooling. In the end, a convolution (1D) block restores the original size of the input, so the reconstructed input is obtained.

### 3.1. Encoding Side

Even if it is not appropriate to talk about encoding and decoding sides in U-net, in this work, the downward and upward sides is also indicated as "encoding" and "decoding", respectively, taking inspiration from the terminology of autoencoders.

### 3.2. Down-Layer and Average Pooling

In this section, a deeper look into the down-layer and average pooling layer is given. Figure 3 shows how the down-layer was created by breaking it out in atomic operations.

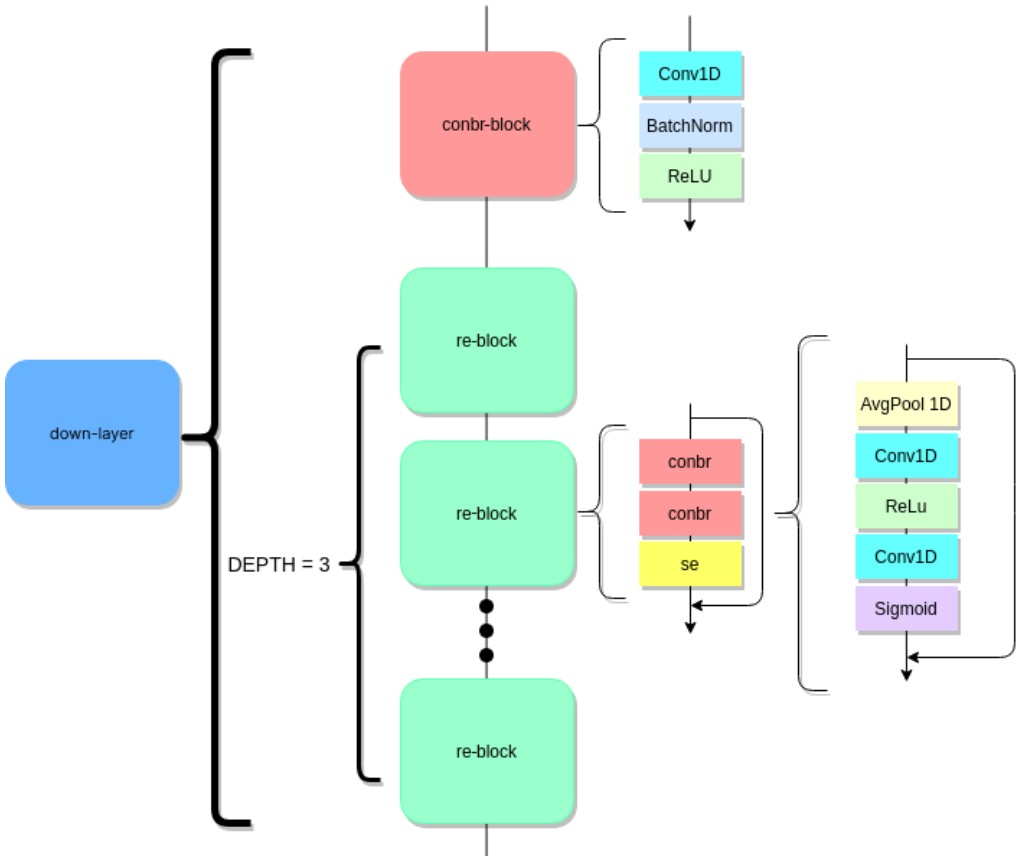

**Figure 3.** A breakdown of the down-layer block.

As can be seen in the figure, the down-layer is composed by two main parts. The first one is the conbr-block (which is the abbreviation of "contraction block"). Conbr-block has the aim of decreasing the number of features of the input. It is not an atomic block as well. During the development of the network we decided to group some atomic blocks for ordering the schema of the whole network: Conv1D, BatchNorm and ReLU.

The second part is composed by a series of re-blocks (number of blocks is handled by "depth" hyperparameter), which concatenate sequentially a conbr-block and a squeeze-and-excitation block. With the squeeze-and-excitation block, the neural networks are better able to map the channel dependency along with access to global information. Therefore they are better able to recalibrate the filter outputs, leading to performance gains. Squeeze and Excitation networks are well explained in Hu et al. (2019).

### 3.3. Decoding Side

The decoding counterpart is a complex block composed by three small pieces: a conbr-block, an upsample block and an attention layer.

### 3.4. Attention Mechanisms in the Proposed Solution

This layer was inspired by Oktay et al. (2018). As the article stated, an attention layer was used in a U-net architecture in order to highlight relevant features passing through the skip connections. The downsampling layer (and, of course, from the underneath up block) gathered information, while the attention layers tried to remove noisy and irrelevant information. This operation was performed before the concatenation operations between the output of the up block and the skip connection result.

Figure 4 shows how the attention block is made up.

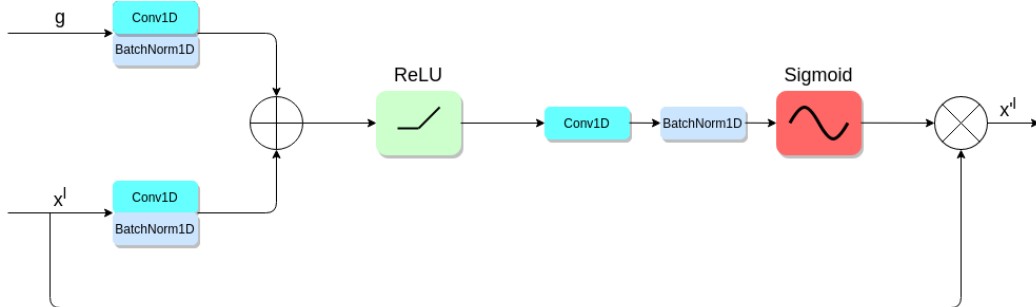

**Figure 4.** Schematic of the proposed additive attention gate (AG).

The block accepts two inputs: the upsampling output ($g$) and the skip connection result ($x^l$). In this case, an additive attention mechanism was proposed. In the aforementioned figure, input features $x^l$ were scaled with the attention coefficients ($\alpha$) computed in the AG. Spatial regions were selected by analysing both the activations and contextual information provided by the gating signal ($g$), which was collected from a coarser scale. A grid resampling of the attention coefficients was performed using a trilinear interpolation. According to Oktay et al. (2018), the result of an additive attention layer is given by the Formula (1).

$$q_{att}^l = \psi^T(\sigma_1(W_x^T x_i^l + W_g^T g_i + b_g)) + b_\psi \tag{1}$$

It is worth to notice that convolutional operations were performed using *kernel* = 1, *stride* = 1 and *padding* = 0, namely a $1 \times 1$ convolution.

### 3.5. Memory Network Block

In this section, we discuss the memory network. This solution was adapted from Park et al. (2020).

A memory network contains M items which store patterns of normal data. A memory network is extremely important because it enables the whole architecture to recognise patterns which are very spread along the time axis.

Items, in memory cells, are denoted by Park et al. (2020) (and hereafter also in this section) as $p_m \in \mathbb{R}^C$ where $m \in [1, ..., M]$. The memory network is able to perform two main operations: reading and updating. Reading means fetching all memory items, performing cosine distance and softmax operation amongst all memory items and input data in order to obtain at the output a feature map which synthesises all normal patterns in the memory network.

Usually, the closest memory item is used to reconstruct patterns. In our case, we considered both the closest one and all the memory items count in the reconstructed feature map computation. Obviously, each read memory item was weighted properly by the usage of the equation previously explained.

The obtained feature map was concatenated with queries and sent as input to the decoder side.

The updating step updated memory items when a new query arrived in the network. Using Equation (2) (paper Park et al. (2020) called this "matching probabilities equation") all closest queries were retrieved, where the $U_t^m$ denote the indices of the corresponding query of the $m$th memory item. The updating rule followed Equation (3).

$$w_t^{k,m} = \frac{exp((p_m)^T q_t^k)}{\sum_{m'=1}^{M} exp((p_{m'})^T q_t^k)} \tag{2}$$

$$p^m \leftarrow f\left(p^m + \sum_{k \in U_t^m} u'^{k,m}_t q_t^k\right) \tag{3}$$

In this case, $f(\cdot)$ represents the L2 norm function. The L2 norm (also known as Euclidean distance) is defined by Formula (4).

$$\|x\|_2 = \sqrt{|a|^2 + |b|^2} \tag{4}$$

As in the reading phase, using a weighted sum instead of a plain sum operation allowed us to better focus on queries which were closest to the memory item.

The computation of $U_t^m$ was performed using a formula similar to (2), as shown in Equation (5).

$$u_t^{k,m} = \frac{exp((p_m)^T q_t^k)}{\sum_{k'=1}^{K} exp((p_m)^T q_t^{k'})} \tag{5}$$

After that, the $U_t^m$ were normalised.

### 3.6. Training Data and Tools

The training and evaluation phases of our algorithm were performed on a remote machine with the following features: Intel Core i7-7820X 3.60 GHz as the CPU, 64 Gb of RAM, 4 × NVIDIA TITAN RTX 24 Gb as GPUs. Particularly, the network used just one of the four GPUs provided by the remote machine.

We used a public dataset to measure the performance of our algorithm; it is available on the Kaggle website Kaggle (n.d.a). It contains transactions made by credit cards in September 2013 by European cardholders. This dataset presents transactions that occurred in two days, with 492 frauds out of 284,807 transactions. The dataset is highly unbalanced, the positive class (frauds) accounts for 0.172% of all transactions. As mentioned in Section 2, imbalance is a key feature of anomaly detection. Generally, dataset imbalance could be addressed using different techniques (as Ali et al. (2019) states), such as random oversampling, random undersampling or SMOTE. The first technique implies samples duplication of the minority class (the class which contains a minor number of samples, in the context of this paper, represented by financial frauds). This approach could overfit the

model. Random oversampling implies samples deletion of the majority class (the class which contains the highest number of samples, in the context of this work, represented by regular banking transactions). This approach has an evident drawback: it could delete important samples and decrease the density of the data in the dataset. The last approach used for addressing imbalanced datasets is SMOTE. This technique aims to generate synthetic samples of the minority class. Generally, it performs better than the first two presented techniques. However, SMOTE emphasises noise propagation of the dataset itself, and an anomaly detection dataset could have a relevant amount of noise due to errors in the data acquisition phase. The approach used in this work was to recognise anomalies using a reconstruction-error-based approach; this technique implied an imbalanced dataset usage.

Given the class imbalance ratio, we measured the accuracy using the area under the precision–recall curve (AUPRC).

Finally, for the validation scenario, we used a pair of Amazon Echo Frames (first generation) smart glasses. We instanced a Web server on an Apple Macbook Pro (processor: 2.3 GHz 8-Core Intel Core i9; memory: 16 GB 2667 MHz DDR4) using Apache Web server v2.4.41. Then, we set up an e-commerce portal using Wordpress 5.7 "Esperanza", with the WooCommerce plugin (version 5.1). As a DBMS, we used MySQL version 8.0.28 installed on the same machine. We additionally installed the Gestpay plugin for WooCommerce, version 20210129, and then we modified it in order to point to a different URL.

Finally we implemented a mock payment processor according to the SOAP WSDL schema of Axerve S.p.A.[1], a company belonging to Banca Sella S.p.A., who provides a payment gateway service. The SOAP protocol version used was 1.2. The fake payment processor application queried the trained algorithm in order to decide if the SCA exemption applied for every payment request arriving from the e-commerce portal.

## 4. Results and Discussion

In this section, we perform and discuss two kinds of validation: a performance validation—which exposes the results of the training of the algorithm over a well-known financial dataset—and a functional validation—about a case study showing a possible use of our AD algorithm to improve the customer experience of digital payments in a smart city.

### 4.1. Performance Validation

We split the dataset in three different subsets: training, validation and test subsets. The training and evaluation subsets were composed of just regular data. This was due to the structure of the architecture and how it used the reconstruction error in order to detect anomalies. The test dataset, on the other hand, was made up by mixed values (regular and anomalous). A validation dataset is a sample of data held back from training the model that is used to give an estimate of the model skill while tuning the model's hyperparameters. The adopted splitting technique divided the dataset using an 80%-10%-10% division. It meant that 80% of the data were used for testing, the remaining 20% were split equally between validation and test slices. The split was not stratified.

### 4.1.1. Hyperparameter Optimisation

In this section, we discuss how the architecture's hyperparameters were tuned properly in order to achieve best performances.

The network presented several hyperparameters that should be finely tuned; however, fitting hyperparameters is very time-consuming, hence we submitted just three hyperparameters to the grid search technique. Grid search performed on multiple parameters took too long and the various training operations were too time-consuming; sometimes training operations overflowed the GPU's capabilities. A grid search is a method of hyperparameters optimisation. It is an exhaustive search of the best result passing through all kinds of combinations. The selected hyperparameters were: learning rate, number of hidden layers of the network and number of memory items.

The learning rate was varied through these values $lr \in 0.001, 0.0001, 0.00001$. This set was chosen because during various attempts it was found that the network performed better under a relatively low learning rate. The number of hidden layers varied as $h\_l \in 32, 64, 128, 256, 512$. This set was made up of elements which were power of 2 (the reason for this lies in the architecture and its convolutional filters). In addition, this set spread from relatively low values to very high values trying to overfit (or better, trying to get closer to the overfitting boundary of) the model. The third hyperparameter concerned the number of memory items which fell in the set $memory\_items \in 2, 10, 20, 50, 100$. This set embraced several orders of magnitude of number of memory items. At the end of the grid search, it should be clear which order of magnitude should be chosen to obtain the best performance of the network. In this regard, the range of values of memory items was augmented after the first phase of the grid search. In fact, a second tuning using this approach was executed in a range of memory items which was much finer than in the first run.

### 4.1.2. Experimental Results

Due to the grid search, several training processes were performed. Table 2 illustrates different AUC score values (expressed in percentage) according to different hyperparameters configurations. The percentage reported in the cells represents the AUC score of the network for that specific configuration.

The training phase was performed using 200 epochs with early stopping. Early stopping prevents a training error curve's plateau. In fact, after some iterations in which the training loss was almost the same (the number of iterations in which there is no improvement is called "patience"), the network was not learning any more. Therefore, the training phase could be stopped before reaching the limit set as a hyperparameter. The evaluation phase (based on the test subset) evaluated all the models previously obtained during the training phase and gave as output the AUC score, precision and recall.

**Table 2.** Scores related to different configuration of two hyperparameters (H = hidden layers, M = memory items).

| H \ M | 2 | 5 | 10 | 15 | 20 | 50 | 100 |
|---|---|---|---|---|---|---|---|
| 32 | 88.68% | 90.16% | 84.42% | 88.26% | 87.54% | 89.70% | 88.12% |
| 64 | 81.00% | 89.29% | 89.58% | 90.06% | 87.85% | 84.82% | 85.14% |
| 128 | 82.45% | 85.24% | 94.27% | 96.08% | 85.86% | 87.54% | 84.13% |
| 256 | 83.18% | 79.46% | 83.18% | 80.69% | 78.51% | **97.71**% | 91.87% |
| 512 | 82.10% | 78.46% | 81.11% | 78.12% | 76.24% | 94.98% | 89.98% |

The sequence of Figure 5 represents the most significant separation histograms of different hyperparameters configurations. The red rectangle refers to the best configuration of hyperparameters. Separation histograms are built using the reconstruction error, that is, the distance between prediction and ground truth. Using this plot, it is possible to appreciate if a network is clearly discriminating regular from abnormal data. The more the Gaussian functions (blue and orange) are separated, the better the network is distinguishing anomaly from normal information.

In the figure, histograms are arranged in order, starting from the smallest configuration (minimum value of hidden layers, learning rate and memory items which was, respectively, 32, 0.001 and 2) and increases the number of hidden layers, memory items and learning rates.

The x-axis plots a numerical reference which helps to appreciate the separation between the histograms, while the y-axis represents the frequency of these data. As mentioned in the legend, the blue bars report normal data and the orange bars report anomalies.

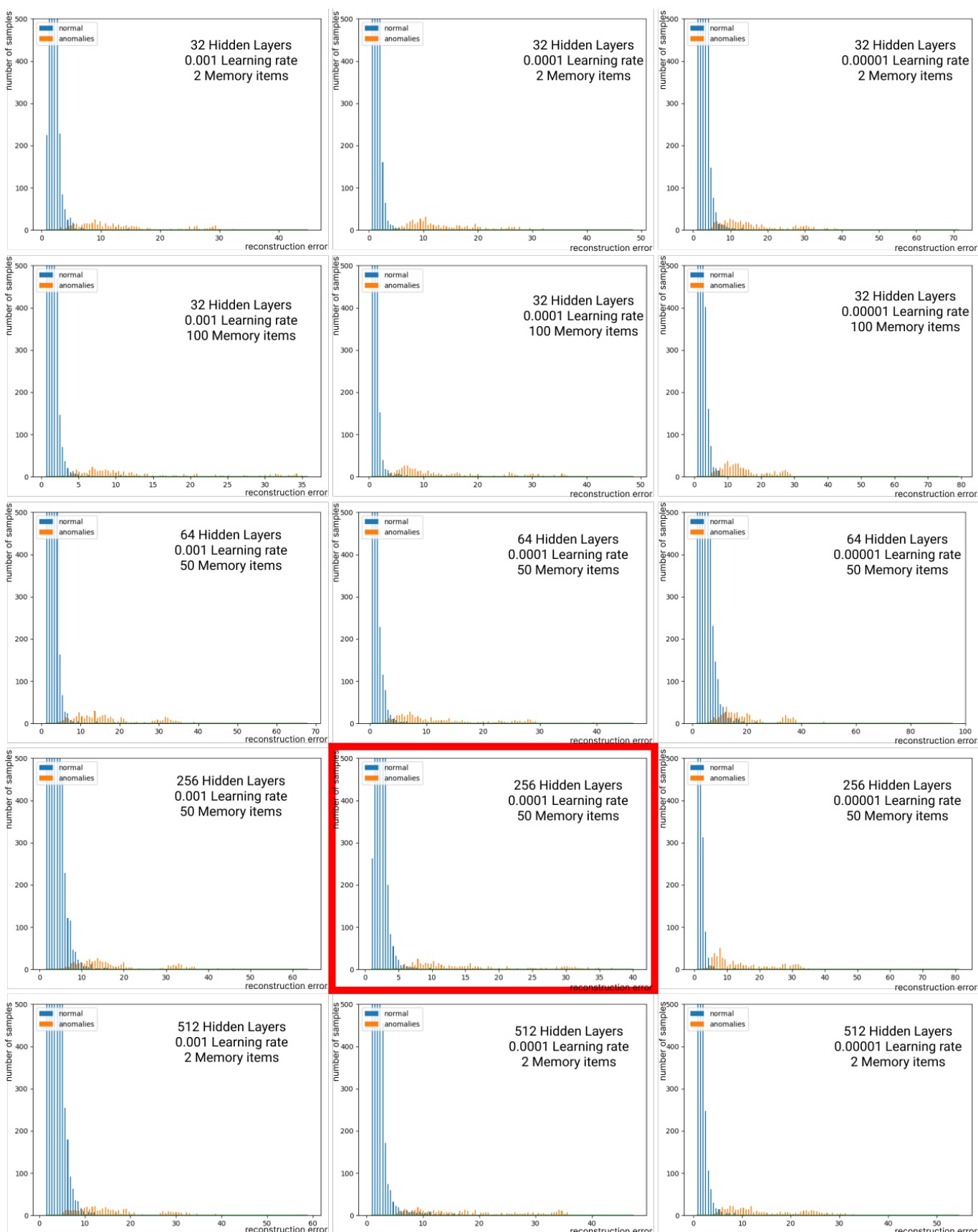

**Figure 5.** Separation histograms of different hyperparameters configurations. The red box refers to the best configuration of hyperparameters.

### 4.1.3. Final Results

After all the training phases, the best set of hyperparameters is evident in Figures 6–8. Actually, the plots depict how AUC scores vary according to, respectively, number of hidden layers, memory items and learning rates, meaning each curve represents the AUC score when other hyperparameters, except hidden layers/memory/learning rates by turns, change.

The best hyperparameter configuration was given by taking the maximum value from all of the three plots. The best model configuration was with 256 hidden layers, 50 memory items and a learning rate equal to 0.001. Bigger values increased the complexity of the network but did not improve its performance.

A series of graphs (Figures 6–8) were derived from each iteration of the grid search method. In particular, the three graphs reported in this work are those that correspond exactly to the maximum values which generated the best model.

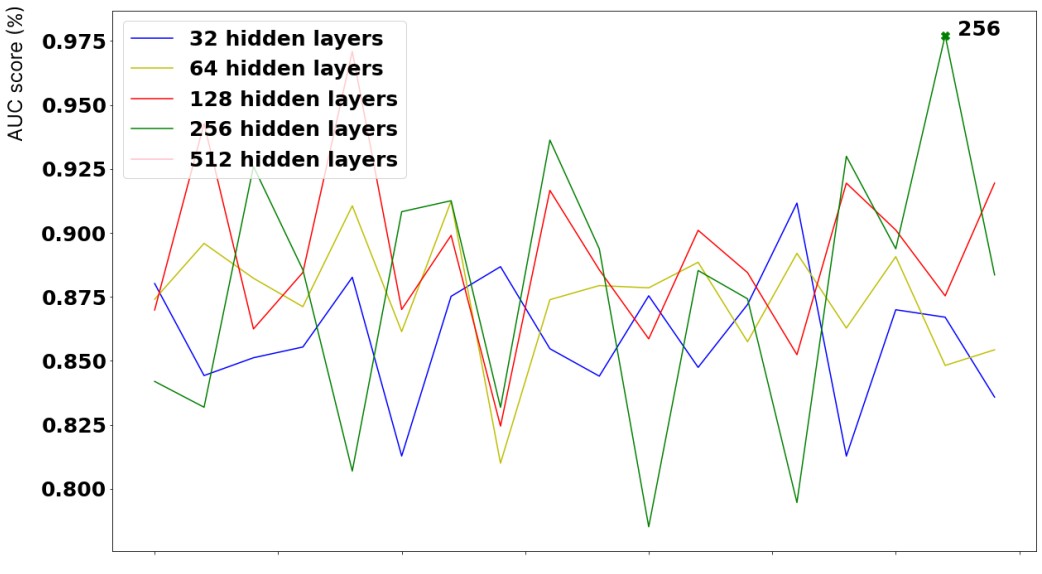

**Figure 6.** Plot showing how the AUC score varies according to different hidden layers.

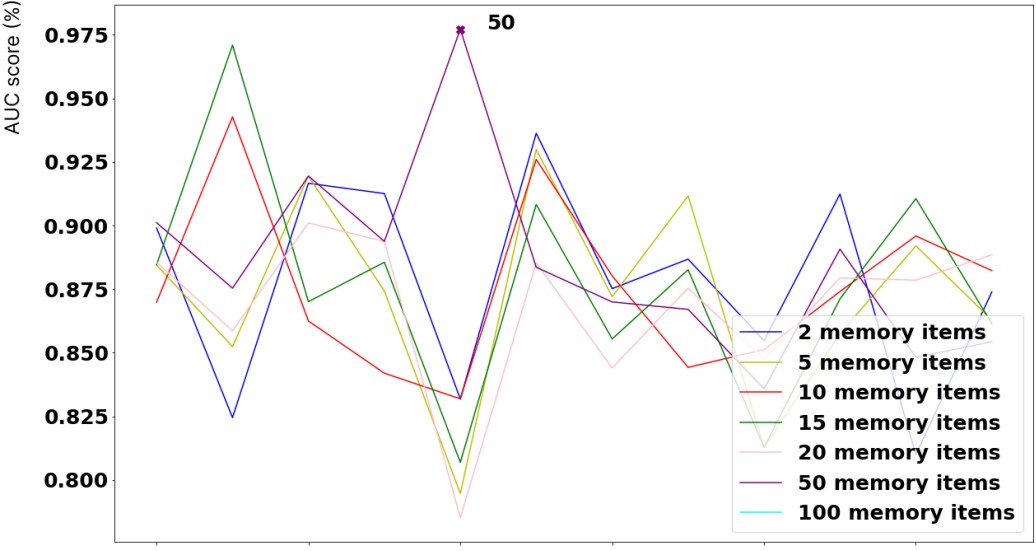

**Figure 7.** Plot showing how the AUC score varies according to different memory items.

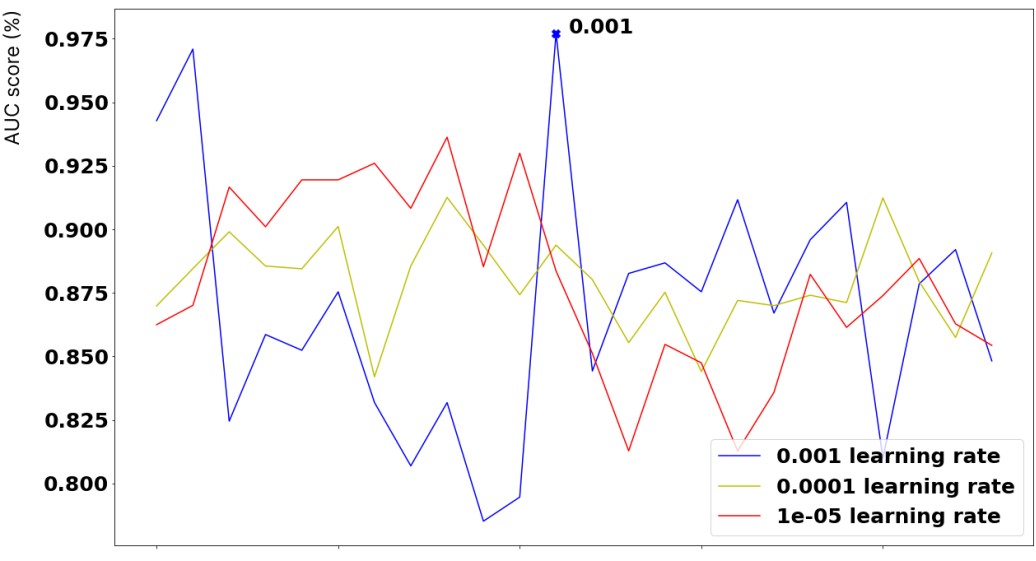

**Figure 8.** Plot showing how the AUC score varies according to different learning rates.

For this reason, the learning curves, separation histogram and confusion matrix relative to the best model are represented in Figures 9–11.

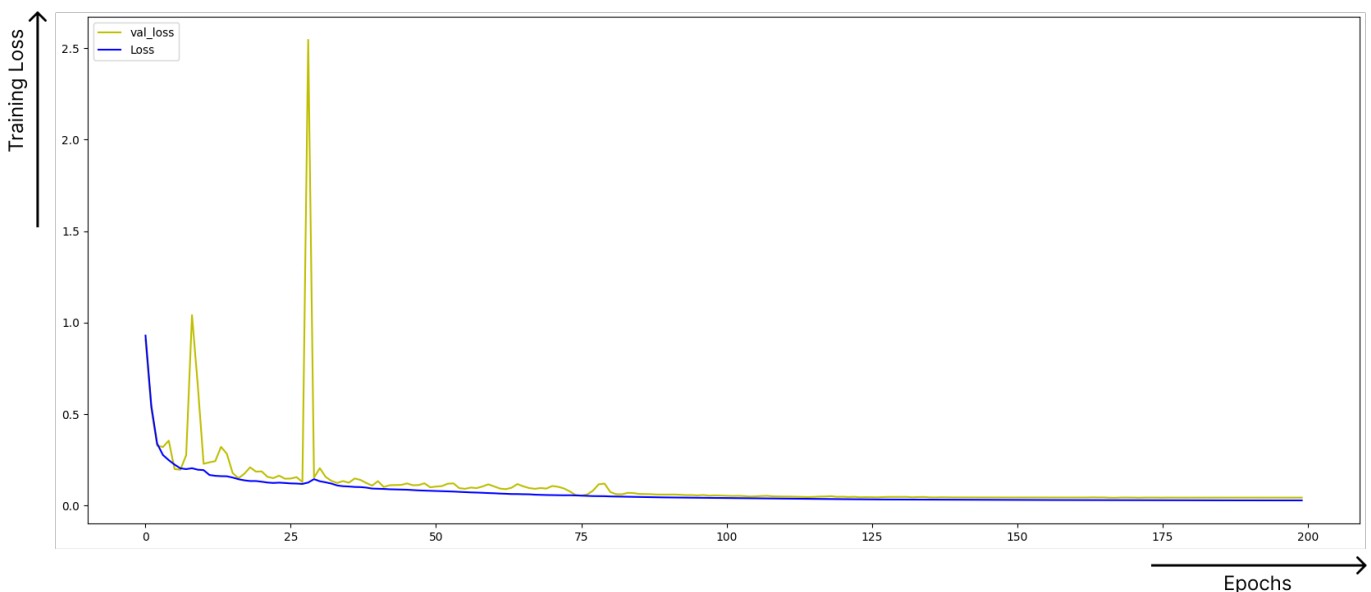

**Figure 9.** Learning curve of the final model after hyperparameter optimisation.

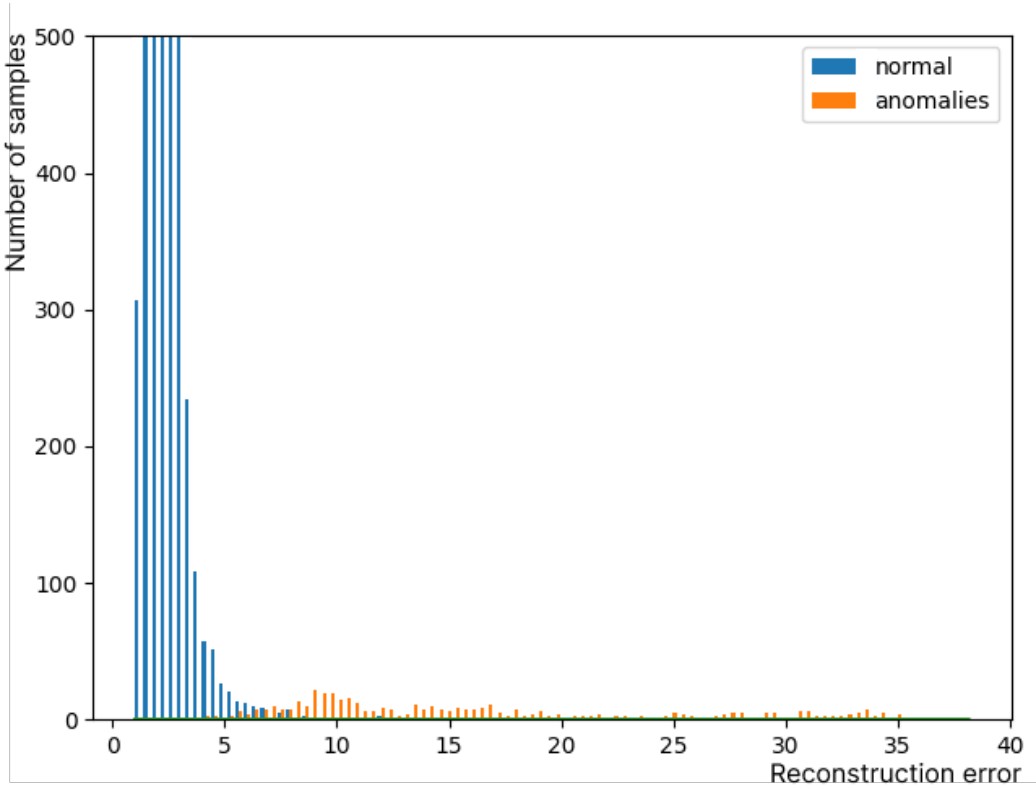

**Figure 10.** Separation histogram of the final model after hyperparameter optimisation.

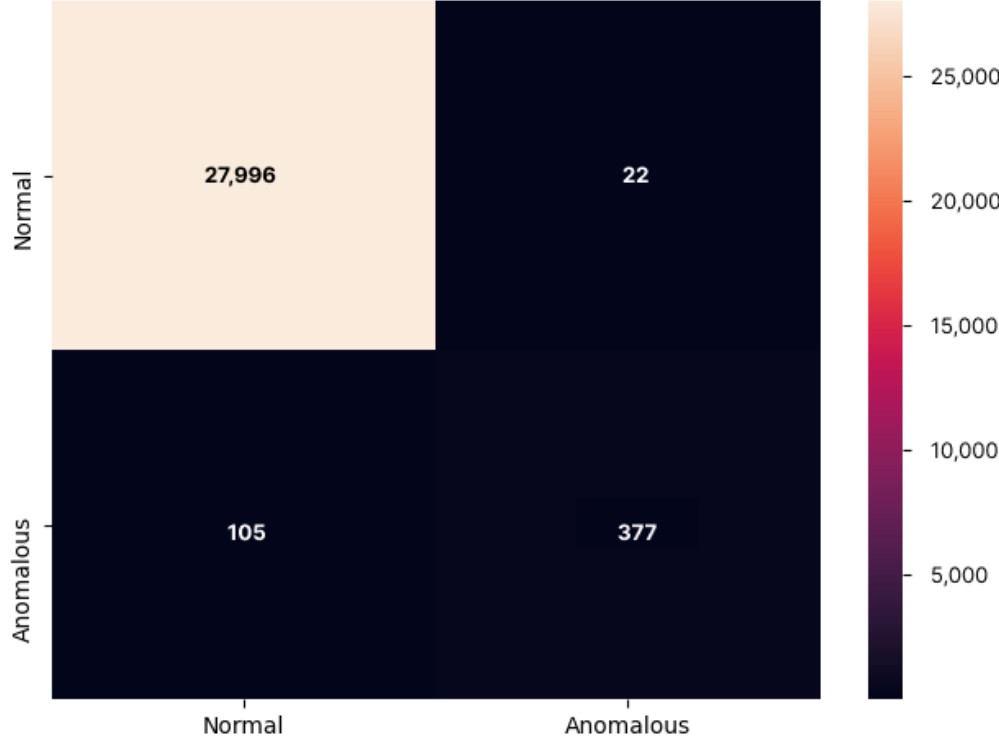

**Figure 11.** Confusion matrix of the final model after hyperparameter optimisation.

In addition to the final model plots, we performed other comparisons in order to evaluate the performance of the architecture. The first comparison was between the proposed solution and the solutions published in the Kaggle's challenge. Table 3 reports the

comparison. It can be noticed that several solutions match other state-of-the-art approaches presented in Section 3.

**Table 3.** A comparison among the proposed solution and other solutions published on Kaggle's leader board.

| Method | Score | Author |
|---|---|---|
| XGBoost | 96% | Kaggle (n.d.c) |
| Random forest | 94.4% | Kaggle (n.d.e) |
| Autoencoder | 94% | Kaggle (n.d.f) |
| Neural network | 92.2% | Kaggle (n.d.d) |
| Decision tree | 87% | Kaggle (n.d.d) |
| Naive Bayes | 90.7% | Kaggle (n.d.d) |
| Nearest neighbours | 96% | Kaggle (n.d.b) |
| **Proposed solution** | **97.7%** | |

In Figure 9, some peaks are visible. These peaks are due to some noncompliant data that should be handled using a preprocessing pipeline, before submitting the dataset to the whole neural network. However, this behaviour does not affect the final result, which looks very smooth.

Moreover, in order to evaluate if the whole architecture benefited from increasing the complexity of the model, a comparison between the final model (merely, U-net, memory network and attention mechanisms) and just the U-net backbone was performed and is reported in Table 4. We found that the memory networks and attention mechanisms really improved the separation ability of the network, as the AUC score was very different in the two alternatives.

**Table 4.** Performances comparison between the complete architecture and the architecture based only on the U-net backbone.

| Hidden Layers | Complete Model (AUC%) | U-Net (AUC%) |
|---|---|---|
| 32 | 89.70% | 86.88% |
| 64 | 84.82% | 88.64% |
| 128 | 87.54% | 77.46% |
| 256 | 97.71% | 76.21% |

### 4.1.4. Comparing Results with State of the Art

Comparing our final results with the scores of other similar works should be done carefully. First, the performance results of the deep learning and traditional machine learning techniques are not comparable, due to the former clearly outperforming the latter due to the great recent advancements in the latest years, especially in treating datasets composed of a huge quantity of data. Moreover, we used only one comparative experimental dataset, even if very representative and used in a significant number of related works. Finally, in the literature, there are several metrics that are able to measure machine learning algorithms' performance, but AD problems require specific metrics for their evaluations, because some classical metrics could be misleading. In AD, it is preferred to not use accuracy as a reference metric due to the accuracy paradox phenomena. A stable metric is the AUC score. A suitable score is represented by F1-score, but the ROC-AUC score is preferred when both classes are equally important for the final classification.

Table 5 reports several solutions based on the same dataset used in this paper. The first column of the table reports the method used to solve the problem. The second to fourth columns contain precision, accuracy, F1-score and AUC score, respectively. In the end, the last column reports the reference for each work.

**Table 5.** Performance comparison between the proposed solution and other solutions based on the same dataset.

| Method | Precision | Accuracy | F1-Score | AUC | Reference |
|---|---|---|---|---|---|
| Logistic regression | 58.82% | 97.46% | 91.84% | N.D. | Varmedja et al. (2019) |
| Naive Bayes | 16.17% | 82.65% | 99.23% | N.D. | Varmedja et al. (2019) |
| Random forest | 96.38% | 81.63% | 99.96% | N.D. | Varmedja et al. (2019) |
| Multilayer perceptron | 79.21% | 81.63% | 99.93% | N.D. | Varmedja et al. (2019) |
| SSO-ANN | N.D. | 93.20% | 95.21% | N.D. | Arun and Venkatachalapathy (2020) |
| Decision tree | N.D. | 71.20% | 80.41% | N.D. | Arun and Venkatachalapathy (2020) |
| Isolation Forest | N.D. | N.D. | N.D. | 95.24% | Porwal and Mukund (2019) |
| Ensemble | N.D. | N.D. | N.D. | 93.11% | Porwal and Mukund (2019) |
| **Proposed solution** | **99.92%** | **99.55%** | **99.77%** | **97.70%** | |

As a final remark of this performance validation, we note that our unsupervised algorithm performs better than the supervised baseline. It is a very strange fact that is not easily explainable, because of the nature of deep neural networks. This paradox is probably dependent on the particular dataset we used.

### 4.2. Functional Validation

This section describes a functional validation we executed to prove the effectiveness of our AD algorithm to deliver innovative and hands-free payments with a high UX. It consisted in developing an e-commerce portal and changing the payment gateway URL so that it pointed to a mock payment gateway connected to an instance of the algorithm.

Figure 12 shows the *in-the-large* architecture of the developed use case. We instanced an e-commerce portal using WordPress and its plugin WooCommerce. Then, we installed the Axerve WooCommerce plugin and we modified it in order to point to a mock payment server which relied on our trained algorithm for performing a transaction risk analysis. If the payment request is typical for the buyer, then the payment process can exempt the SCA.

When we cite HF-SCA, we refer to the whole novel payment scenario leveraging an instance of the AD algorithm exposed in the previous sections. In HF-SCA, the AD algorithm is used as a transaction risk analysis tool, whose aim is not to detect frauds but to improve the customer experience of digital payments by detecting regular payments that could benefit from the SCA exemption.

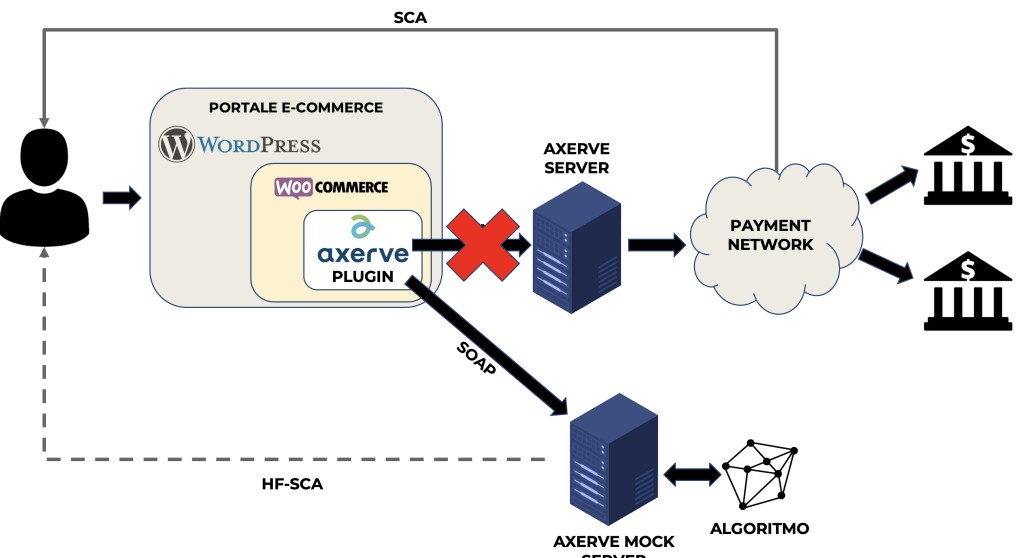

**Figure 12.** Architecture of the case study.

What we wanted to achieve with this case study was the empirical demonstration of two main benefits of HF-SCA:

1. HF-SCA can deliver a much shorter customer journey, hence bypassing (when the risk is low) the need for SCA;
2. HF-SCA enables innovative use cases, because customers have their hands free to perform other activities while making online purchases.

With regards to the first benefit, in Figure 13 we show a comparison of the customer journey between an online purchase using classical SCA to authorise payments and the refactored version using the HF-SCA algorithm to seamlessly authorise payments. The schema shows what we expect in terms of improvement of the customer's payment journey. Being much shorter, the flow on the right side is much less error-prone, both from the customer side—who enjoys the SCA exemption for regular purchases—and the system side—because of technical errors that can arise during the SCA process.

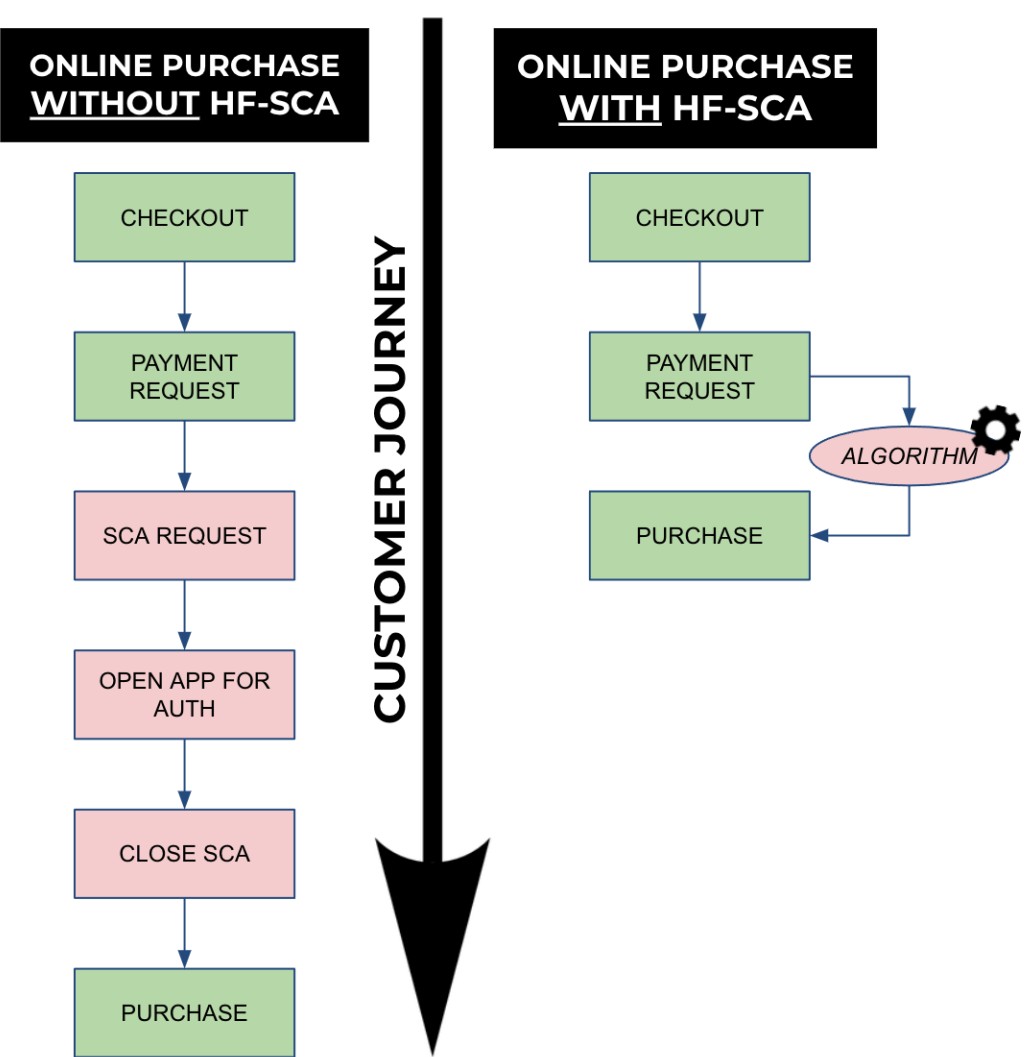

**Figure 13.** Comparison of the customer journey between classical SCA and HF-SCA algorithm in online purchases.

In order to experiment the second benefit, we made the e-commerce portal available both via a traditional web browser and also with an Amazon Alexa skill, which was deployed in the AWS Cloud by using both Amazon Skills Kit (ASK) service and AWS Lambda. The vocal experience best benefits from the advantages derived by HF-SCA, because it allows the delivery of payment use cases completely hands-free (no need for the customer

to shift their attention to a second factor such as a mobile phone). Delivering hands-free payment service can support several scenarios in a smart city, such as the possibility for the customer to settle payments while driving. Smart mobility is a component of the smart city concept, where the acknowledged management of enterprises' relationships with customers within their service is profoundly determined by the deployment of engaging applications.

The choice of this particular scenario was driven by the prior expertise of some of the authors in the Internet of Things field Caione et al. (2017). In Figure 14, we show the Amazon Echo Frames used for the experimentation, and a picture of the tester driving their car while making hands-free purchases via our skill.

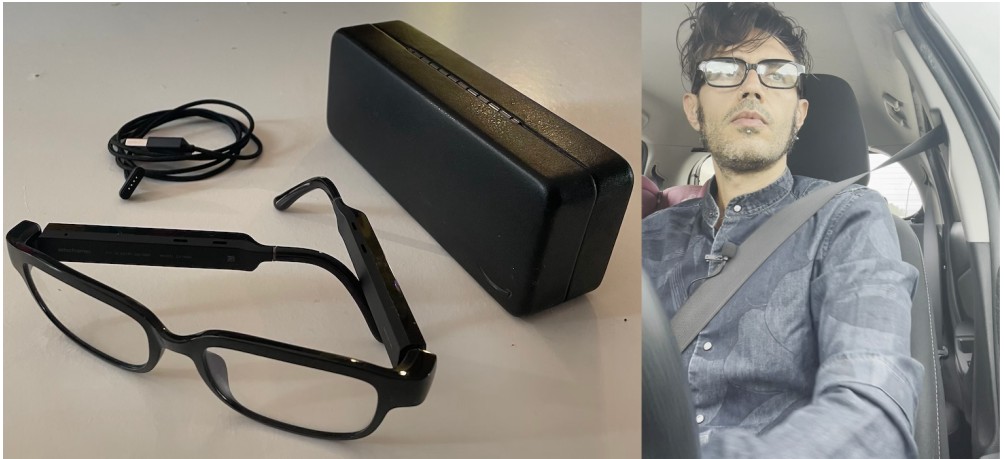

**Figure 14.** Using Amazon Echo Frames smart glasses (on the left) to make a secure online purchase while driving.

While the architecture of this case study presents no technical novelty, to the best of our knowledge it is the first test bed in the literature exploiting SCA exemptions to deliver a high UX in digital payments for a smart city. While TRA tools are commonly used to detect anomalous transactions (end hence stop or flag them), in our inverse approach, we help to identify regular cases in which the two-factor authentication can be safely bypassed, and in such cases engaging and hands-free payment interactions can be set up.

## 5. Discussion

Inspired by a challenge Sella (n.d.) organised by Banca Sella S.p.A., an Italian private bank, we developed a deep-learning-based anomaly detection neural network. We combined existing ML techniques to create a novel approach to AD in the financial context. Our algorithm has a neural network based on a U-net architecture as a backbone, augmented by a memory network and an attention mechanism to improve the network's separation ability.

We used a public dataset retrieved online Kaggle (n.d.a) on the Kaggle website. The AUC score was the reference metric used for comparing the performance of the proposed network in the Kaggle's leaderboard. Using this approach, a 97.7% AUC score was obtained. To the best of our knowledge, our algorithm—which we called HF-SCA (hands-free strong customer authentication)—outperformed other AD approaches in the literature using the same public dataset. Another important finding is that 98% of transactions could be executed securely as exemptions to the SCA process, because of a low risk rightly guessed by the algorithm.

To sum up, the network benefits by three main concepts implemented in it:

- Autoencoder. Despite the main backbone not being a proper autoencoder, U-net is based on the same idea. It adds skip connections which provide an alternative path to gradient (with backpropagation) avoiding the gradient-vanishing phenomenon. In

addition, a substitution of standard convolution blocks was made, replacing them with squeeze-and-excitation blocks. They enhance the representation ability of the network and perform dynamic channel-wise features recalibration.

- Attention mechanisms. They facilitate fraudulent patterns recognition by the network. The whole architecture tries to focus its attention on some features. The attention logic is led by these mechanisms.
- Memory network. It lies on the latent space, replacing the bottleneck layer introduced by the original U-net architecture. The memory layer boosts the AD ability, plunging the architecture in a multimodal environment.

Two points of attention to be considered when training and deploying HF-SCA are the following.

The first is related with the flexibility of the network in term of transactions. In our approach, a sliding window of 500 transactions was used (and this aspect could represent a constraint for datasets poorer than the one we used). Due to the hyperparameters characterising the convolutional layers, currently it is not possible to go under this value. Some attempts have been already made about this adjustment and it seems to not be so difficult to implement sliding windows smaller than 500.

The second is the number of features which the algorithm supports. Currently, the dataset has to have a number of descriptors equal to a power of two, this is clearly an industrial constraint. In contrast to this drawback, based on our expertise coming from our relationships in the banking field, a number of 32 features is a common upper threshold for datasets in the financial context. This is probably due to the typology of data which are reported in these datasets. On the other hand, resolving this drawback implies deleting padding in order to make the dataset, in terms of a number of features, suitable for this network.

## 6. Conclusions

In this work, we presented a novel transaction risk analysis tool that can be used to trigger SCA exemptions in digital payments. The SCA process is fundamental to increase the security of digital payments. Nevertheless, the complexity of the two-factor authentication led to a 22% increase in e-carts abandonment in the first semester of 2021. The main customer segment who suffered from this new duty is the one including the less accustomed to technology, mainly older generations, but also those customers who simply cannot afford a second device. This goes in the opposite direction of the aims of the smart city, whose manifesto calls for creating and fostering accessible urban services for everyone.

As an example of usage of our algorithm in a smart city, we presented a case study in which a tester used a pair of Amazon Alexa smart glasses to settle payments to an e-commerce portal by simply using their voice, without distraction due to the SCA process. The user was able to buy products while driving their car, without the need for using a smartphone, whose use could be very dangerous while driving.

As a next step, we are now working on training the HF-SCA algorithm on a private dataset provided by our partner Banca Sella. The project was also selected by Bank of Italy to be part of the first national regulatory sandbox in Italy Bank of Italy (n.d.). Future research efforts will include the comparison of the performance of the same algorithm fed by different datasets.

**Author Contributions:** Conceptualization, R.V.; methodology, C.D. and L.M. (Luigi Manco); software, B.T. and L.M. (Luigi Manco); validation, L.F.; formal analysis, L.M. (Luigi Manco); investigation, R.V.; resources, L.M. (Luigi Manco) and L.F.; data curation, B.T.; writing—original draft preparation, R.V. and B.T.; writing—review and editing, R.V.; visualization, R.V. and B.T.; supervision, L.M. (Luca Mainetti); project administration, R.V. All authors have read and agreed to the published version of the manuscript.

**Funding:** This research received no external funding.

**Institutional Review Board Statement:** Not applicable.

**Informed Consent Statement:** Not applicable.

**Data Availability Statement:** We used a public dataset to measure the performance of our algorithm. It is available on the Kaggle website Kaggle (n.d.a). It contains transactions made by credit cards in September 2013 by European cardholders.

**Conflicts of Interest:** The authors declare no conflict of interest.

## Note

1       https://ecomms2s.sella.it/gestpay/GestPayWS/WsCryptDecrypt.asmx?wsdl, accessed on 24 July 2022.

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
