# Peer review of "HF-SCA: Hands-Free Strong Customer Authentication Based on a Memory-Guided Attention Mechanisms"

_jrfm, doi:10.3390/jrfm15080342_

Round 1

Reviewer 1 Report

I strongly suggest that an English native speaker proofreads the paper.

The authors must check references. Sometimes they have "'?" inside brackets, i.e. [?].

Why did the authors not use some known technique to address the imbalanced dataset?

The authors should provide more details on how they split the dataset. Was the split stratified?

Otherwise, the paper is well-written. The introduction well presents the problem of the paper and its importance. The literature review is sufficient. The proposed algorithm is well defined, and readers can easily follow the explanation.

Used references are relevant and up-to-date.

Author Response

I strongly suggest that an English native speaker proofreads the paper.

Thank you for your suggestion. Due to lack of time, if this review is ok, we kindly ask some more time in order to proofread the manuscript.

The authors must check references. Sometimes they have "'?" inside brackets, i.e. [?].

We carefully checked the references and we fixed the broken citation "?".

Why did the authors not use some known technique to address the imbalanced dataset?

As better clarified on the paper, the approach used in this work is to recognize anomalies using a reconstruction error-based approach, this techniques implies an imbalanced dataset usage.

The authors should provide more details on how they split the dataset. Was the split stratified?

As better clarified in the paper, the adopted splitting technique divides the dataset using an 80%-10%-10% division. It means that the 80% of data is used for testing, the remaining 20% is split equally between validation and test slices. The split was not stratified.

Otherwise, the paper is well-written. The introduction well presents the problem of the paper and its importance. The literature review is sufficient. The proposed algorithm is well defined, and readers can easily follow the explanation.

Thank you very much!

Used references are relevant and up-to-date.

Thank you!

Reviewer 2 Report

I would suggest authors address the following concerns before publication: 

1) Figure 5 shows the histograms of different hyperparameter configurations. Please add the axis titles. 

2) Figure 6, 7 and 8 show the AUC scores, please also add the axis label and unit to avoid confusion 

3)  Please include the axis title and unit in Figure 9. Can the authors explain the 2 peaks on the curve

4) Please include the axis title and unit for figure 10, and move the legend box to top-right

5) 3 out of 4 subfigures in figure 11 are similar, which are hard to view. Even though their numbers are close, is there a way to change the type of display to show the trend? 

Author Response

1) Figure 5 shows the histograms of different hyperparameter configurations. Please add the axis titles. 

We added the axis titles in figure 5. Thank you.

2) Figure 6, 7 and 8 show the AUC scores, please also add the axis label and unit to avoid confusion 

We added them in the figures.

3)  Please include the axis title and unit in Figure 9. Can the authors explain the 2 peaks on the curve

We added the axis labels. As clarified in the paper, peaks are due to some non-compliant data. However, this behaviour does not affect the final result that looks very smooth.

4) Please include the axis title and unit for figure 10, and move the legend box to top-right

This was done. Thank you!

5) 3 out of 4 subfigures in figure 11 are similar, which are hard to view. Even though their numbers are close, is there a way to change the type of display to show the trend? 

Unfortunately, due to their similar values, the colours of the 3 cells in the confusion matrix are very similar. We can't change their colours without altering  the meaning behind the heat map. We are very sorry about that.

Reviewer 3 Report

- The Literature Review chapter is definitely not „in-depth overview” as the authors indicated in the Introduction; it is a very shallow review which does not include many basic terms for this paper, e.g. more information about described research of SCA, HF-SCA, PSD2 should be added; moreover, there is lack of explanation who is the customer in the authors’ assumptions – please add,

- Fig. 1 needs more explanation: What do the colours mean? What do the numbers mean? Why author chose U-net architecture to present in the figure while the other solutions for AD are ignored?

- The title of Figure 2 is only partial – of which architecture?

- The titles of Fig. 3- 14 are  wrong – they are rather descriptions, not the titles – please correct them,

- The architecture of the case study is completely standard with no novelty  - what is the reason to present it?

- There is a lack of a Discussion chapter – please add.

Author Response

- The Literature Review chapter is definitely not „in-depth overview” as the authors indicated in the Introduction; it is a very shallow review which does not include many basic terms for this paper, e.g. more information about described research of SCA, HF-SCA, PSD2 should be added; moreover, there is lack of explanation who is the customer in the authors’ assumptions – please add,

Thank you for pointing it out. We added a literature review about PSD2 and SCA at the beginning of Section 2. Moreover we better described who is the customer of the system in the introduction (the users who make online card-not-present transactions).

- Fig. 1 needs more explanation: What do the colours mean? What do the numbers mean? Why author chose U-net architecture to present in the figure while the other solutions for AD are ignored?

We added an explanation about the colours and the numbers near the picture, thank you. U-net allows to perform anomaly detection using a reconstruction error-based method in a similar way to autoencoders. However, due to deep levels of the network itself, skip connections allow avoiding gradient-vanishing effect and thus having better performance during training and evaluation phases. We placed this additional explanation in section 3.

- The title of Figure 2 is only partial – of which architecture?

We better clarified the title of Figure 2. Moreover, we added some extra description just below the picture. Thank you!

- The titles of Fig. 3- 14 are  wrong – they are rather descriptions, not the titles – please correct them,

We fixed all the titles in the 3-14 interval, and we moved the former descriptions in the text, near the related pictures. Thanks.

- The architecture of the case study is completely standard with no novelty  - what is the reason to present it?

While the architecture of this case study presents no technical novelty, to the best of our knowledge it is the first testbed in the literature exploiting SCA exemptions to deliver high UX in digital payments for the smart city. We added this explanation at the end of section 4.

- There is a lack of a Discussion chapter – please add.

Since the conclusions sections was too big (as containing the discussion themself), we extracted the discussion and moved it a separate chapter (section 5), modifying the 2 texts in order them to be sound. Thanks.

Round 2

Reviewer 3 Report

The authors made the changes in the paper, however, there still are two important areas which need corrections:

- the remark which was addressed to the titles of the figures is also adequate to the titles of the tables  – they are rather descriptions, not the titles – please correct them,

- the relation between customers, applications and smart city is not introduced in the literature chapter or introduction; please explain them using  the reference:

 KadÅ‚ubek, M.; Thalassinos, E.; DomagaÅ‚a, J.; Grabowska, S.; Saniuk, S. Intelligent Transportation System Applications and Logistics Resources for Logistics Customer Service in Road Freight Transport Enterprises. Energies 2022, 15, 4668. https://doi.org/10.3390/en15134668

Author Response

- the remark which was addressed to the titles of the figures is also adequate to the titles of the tables  – they are rather descriptions, not the titles – please correct them,

We adjusted the titles also for tables. Thank you!

- the relation between customers, applications and smart city is not introduced in the literature chapter or introduction; please explain them using  the reference:

This has been better specified by using some concepts of the provided reference, and obviously citing it. Thank you!

Round 3

Reviewer 3 Report

The authors corrected the paper except for the title of Table 2 - please also make the correction there, the same as in other tables...

Author Response

Very sorry, it slipped out. Many thanks.